# Sustainable International Relations. Pope Francis' Encyclical *Laudato Si'* and the Planetary Implications of "Integral Ecology"

**Pasquale Ferrara** [1,2] 

1   Diplomacy and Negotiation, LUISS Guido Carli, 00197 Roma, Italy; pferrara@luiss.it
2   International Relations and Integration, Sophia University Institute, 50064 Florence, Italy

**Abstract:** This paper analyzes the theoretical and pragmatic implications for international relations and world politics of the new holistic approach to climate change articulated by Pope Francis in the Encyclical *Laudato Si'*, particularly through the notion of "integral ecology". It is not my intention to offer an exegesis of the Papal document. I will rather try to illustrate and discuss its planetary hermeneutics. I emphasize that the Encyclical's perspective is not exclusively normative, and that, within the dynamic interplay between social structure and human agency, it can also be considered as a call to action. In this context, I suggest that both International Relations Theory and global politics have much to learn from the fundamental claims of contemporary religions in relation to climate disruption. In particular, Pope Francis' document, far from being just a new chapter in the unfolding process of the "greening" of religions, raises the issue of the sustainability of the present world system. Therefore, I contend that the perspective of the Encyclical calls for a radical transformation of international relations, since it emphasizes the deep implications of environmental issues on the entire spectrum of security, development, economic and ethical challenges of contemporary world politics. Against this backdrop, my objective is to connect the main tenets of the Encyclical to the environmental turn in International Relations Theory and to the new epistemological challenges related to the paradigm shift induced by the new planetary condition of the Anthropocene and the relevant questions arising for a justice encompassing the humanity-earth system. The Encyclical seems to suggest that practicing sustainable international relations means exiting the logic of power or hegemony, while simultaneously operationalizing the concept of care.

**Keywords:** climate change and international relations; religion and the environment; Anthropocene; planetary justice; sustainability

## 1. Introduction

"The present world system is certainly unsustainable" (LS[1], §61). This striking statement appears in Pope Francis' Encyclical *Laudato Si'* (Francis 2015). It is my intention to take seriously this sharp evaluation of the current state of international relations in order to understand, without pretending to present an exclusive interpretation, what it really entails, from both a theoretical and a practical perspective.

In this paper I will not, however, try to offer an exegesis of the Papal document. Rather, I will concentrate on its hermeneutical focus as a "comprehensive vision" of the present planetary challenge.

---

1   In this article Pope Francis' Encyclical *Laudato Si'* is quoted, after the initials of the title, as "LS" followed by the number of the relevant paragraph.

I will highlight that the Encyclical's perspective is not exclusively normative. It is also a call to action, comparable, to some extent, to a social and political *manifesto*, within the dynamic interplay between social structure and human agency.

The Encyclical is a radical interpretation of the environmental crisis. Its author is, moreover, a highly influential transnational religious actor (the Holy See), whose international projection can be investigated by applying the notion of "milieu goals" (Wolfers 1962). According to Wolfers, in a state's foreign policy one can distinguish between "possession goals," and "milieu goals."

> In directing its foreign policy toward the attainment of its possession goals, a nation is aiming at the enhancement or the preservation of one or more of the things to which it attaches value. ( . . . .) Milieu goals are of a different character. Nations pursuing them are out not to defend or increase possessions they hold to the exclusion of others, but aim instead at shaping conditions beyond their national boundaries. (Wolfers 1962, pp. 73–74)

In the case of "milieu goals" an actor cannot obtain its goals autonomously. Therefore, in these circumstances, actors try to (re)shape to their advantage the social context in which they act. The "milieu goals" are different from those of "possession", because they are inclusive, distributive, "non-zero sum".

The Holy See is not a State (in a Westphalian way); however, it performs its role in terms of moral agency well beyond the boundaries of the Catholic religious community. Concern for the environment and the corresponding willingness to engage in the attempts to halt the deterioration of "our common home" can be regarded as a "milieu goal" pursued by the Holy See, particularly under Pope Francis' leadership.

More broadly, I suggest that both International Relations Theory and global politics have much to learn from contemporary religion's fundamental claims in relation to climate disruption. These claims include, for example, inclusive concepts such as "universal communion" (see below) as an alternative to the cosmopolitan/communitarian divide, or care for "our common home" as an analytical framework implying direct accountability for the use of the common goods (such as air, water and land).

More precisely, if we were to conceptualize the role of the Catholic Church, in the light of the LS, in relation to the global challenge of climate change, we could perhaps apply the analytical category of "environmental stewardship" (Falkner and Buzan 2017). Environmental stewardship describes the role of relevant international players in acknowledging both the seriousness of the crisis of climate change and taking the responsibility to act (Falkner and Buzan 2017, p. 147). Environmental stewardship has an impact on the foundational institutions of international relations, namely sovereignty and territoriality (Falkner and Buzan 2017, p. 149). For that reason, "environmental stewardship might eventually require deeper levels of solidarist cooperation and institution-building than mere policy coordination" (Falkner and Buzan 2017, p. 149).

## 2. The Socio-Natural Hermeneutics of *Laudato Si'*

The Encyclical's contribution to public discourse is significant from three different perspectives. First, theologically speaking, Pope Francis' approach helps believers to translate their faith convictions into active environmental concerns. Second, philosophically speaking, it allows an "ethical exploration of the relationship between people and nature" by "highlighting the importance of ethical discussion in scientific debate" (O'Neill 2016, p. 752). Third, politically speaking, the Encyclical deals with important issues of public policy, in particular by advocating for a critical judgement of the "dominant technocratic paradigm" and the myth of unlimited growth (O'Neill 2016, p. 750). Pope Francis' underlying discourse can be described as "a recurring call for individuals to take responsibility, for civic groups to maintain political pressure, and for interdisciplinary dialogue about a more ethical and equitable way forward" (O'Neill 2016, p. 754).

My preliminary argument is that the Encyclical deals less with environmental issues than with a political philosophy of world politics. More than this, it is a cosmological appraisal of the human

condition on the planet. Pope Francis refers explicitly to "a strategy for real change" that requires "rethinking processes in their entirety, for it is not enough to include a few superficial ecological considerations while failing to question the logic which underlies present-day culture" (LS, §197). The ecological crisis is the starting point for a much wider outlook of the declining quality of human life, growing global inequalities, the technological hegemony that marginalizes social concerns. In the document this approach is defined as "integral ecology", meaning a wide range of issues regarding the "relationship existing between nature and the society which lives in it" (LS, §139). The reference to "society" implies not only human communities living within political boundaries, but also the human community as a whole and as the infrastructure of relations between the existing political units on the planet (international relations).

Integral ecology is understood, in the Encyclical, as "a vision capable of taking into account every aspects of the global crisis" (LS, §137). Integral ecology (not to be confused with radical ecology, which implies a neo-Marxist outlook; Merchant 2005) is an innovative concept, comparable to the notion of "integral development" introduced in the political and social discourse of the Catholic Church by Pope Paul VI (O'Neill 2016). In practice, "an integral ecology refers to the fact that everything is interrelated, requiring consideration of the interactions within natural systems and with social systems. Accordingly, the Pope sees a single, complex crisis that is both social and environmental" (O'Neill 2016, p. 750).

This dramatic emphasis on the multi-dimensional crisis of the environment marks a sharp difference vis-à-vis the traditional Catholic discourse on the harmony of the Creation, and turns the Franciscan contemplation of nature into a serious and somewhat divisive political issue at the global level.

In particular, I contend that the Encyclical's perspective has to do with the claim of a radical transformation of international relations, since it emphasizes the deep implications of environmental issues on the entire spectrum of security, development, economic and ethical challenges of contemporary world politics. This holistic perspective seems to echo the authoritative Brundtland Report, a document of utmost importance for the public awareness of the interrelated factors causing environmental degradation:

> the environment does not exist as a sphere separate from human actions, ambitions, and needs, and attempts to defend it in isolation from human concerns have given the very word "environment" a connotation of naivety in some political circles. The word 'development' has also been narrowed by some into a very limited focus, along the lines of 'what poor nations should do to become richer', and thus again is automatically dismissed by many in the international arena as being a concern of specialists, of those involved in questions of "development assistance". (World Commission on Environment and Development 1987, p. 13)

The "environmental turn" in international relations cannot therefore be reduced to a mere normative question regarding ecological responsibility. The issue at stake is the search for a new analytical focus in international relations which is radical, trans-disciplinary, trans-paradigmatic. First and foremost, the centrality of the questions related to the eco-system challenges the concept of exclusivist sovereignty exactly when the latter seems to have become the new normal in world politics. As the Encyclical puts it, "inequity affects not only individual but entire countries; it compels us to consider an ethics of international relations" (LS, §51).

## 3. Religions and the Environmental Crisis

Religions—especially Judaism and Christianity—have long ignored or underestimated the seriousness of the environmental crisis, sometimes as the result of an approximate understanding of biblical passages, such as a narrow interpretation of Genesis 1:26–28, God's command to "rule over" Earth's creatures. The idea of dominion rather than care reduces human responsibility towards

creation to a peculiar and arbitrary legitimacy of anthropocentrism. The notion of power has been the leading paradigm vis-à-vis nature and creation, "asserting that humans are different from, better than, or independent from nature" (Gottlieb 2009). Pope Francis strongly criticizes this anthropocentric "supremacy" thesis or "dominionism", by stressing that "nowadays we must forcefully reject the notion that our being created in God's image and given dominion over the earth justifies absolute domination over other creatures" (LS, §67). The "excessive" anthropocentrism brought about by modernity is also connected to the methodological individualism of social sciences, as long as it "continues to stand in the way of shared understanding and of any effort to strengthen social bonds" (LS, §116). The unbalanced emphasis on the individual influenced also "an inadequate presentation of Christian anthropocentrism" that "gave rise to a wrong understanding of the relationship between human beings and the world" (LS, §116). Pope Francis denounces the "Promethean vision of mastery over the world" (LS, §116), which relegates the protection of nature to allegedly "faint-hearted" people (LS, §116). On the contrary, the "dominion" over the universe "should be understood more properly in the sense of responsible stewardship" (LS, §116). Islamic scholars have recently re-framed the need to respect of the environment as one of the central themes of the Qur'an. In particular, Islamic environmentalists stress that what God entrusted to mankind is not dominance over creation, but rather "stewardship" ("*Khalifah*") (IDoGCC 2015); a concept entailing collective responsibility (Hancock 2018).

In general, in the face of a the growing social-environmental crisis, religions are becoming more aware that "responding to this challenge is profoundly altering every aspect of religious life: theology, institutional self-definition, the everyday conduct of religious people, and rituals" (Gottlieb 2009, p. 492). Particularly interesting is the appearance of new strands of theological studies, namely eco-theology. Equally relevant is the effort to propose a re-interpretation of religious traditions, more in line with prophetic social criticism. In Christianity, "the long-standing tendency to make categorical distinctions between body and spirit, the world and the soul, the moral status of people and the non-moral status of everything else, has been sharply criticized" (Gottlieb 2009, p. 495).

Of utmost importance is also the new attitude of religious authorities, who in some cases have shown a spectacular turn towards a more integrated and holistic vision of creation. Orthodox Patriarch Bartholomew I, for example, in 1997 called any crime committed against the natural world "a sin" (Bartholomew 1997). John Paul II in 2000 talked about "fraternity with the earth" (John Paul 2000). This new approach represents "not only a bold and non-human centered environmental declaration; it is also a ground-breaking move from a religious tradition which for many centuries did its best to destroy any religion who honored the earth" (Gottlieb 2009, p. 497). As an example of this new commitment, the December 2018 issue of the Ecumenical Review the World Council of Churches focuses on the "Theology of the Oikos", in the conviction that "theology can provide solutions for the sustainability issues that challenge the common home of humanity" (WCC 2018). In the protestant camp, the Encyclical *Laudato Si'* has been insofar as it brings about a "paradigm shift from rulers of the environment to partaking in universal fraternity in the common home" (Bals 2016, p. 42).

The interreligious dialogue on climate, justice and sustainability has already started not only in the ecumenical context, but also between global religions, namely Christianity and Islam (Mattson et al. 2011). In Judaism the reflection on the environment is now established and articulated (Tirosh-Samuelson 2002).

Religious communities play a double role vis-à-vis the ecological crisis: on the one hand, they help to frame the discourse on climate change in the broader context of solidarity and equity, as well as on the theological concept of "Creation" as a heritage to be respected and cherished; on the other hand, they have a mobilizing power in the context of campaigns against the unfair exploitation of nature, especially when that risks damaging pre-existing native communities. Moreover, religions may play a crucial role by influencing the decision-making process of governments and international organizations on climate issues (Grape 2016). More fundamentally, religious communities are capable of developing a "hermeneutics of land" as a tool for debating the "religio-cultural motivations to change patterns of consumption, pollution and the means of forming human relations" (Wellman 2010, p. 35).

To be sure, the trend toward the "greening" of religions should not be taken for granted, and special attention should be devoted to sophisticated faith-based approaches challenging the man-made climate change assumptions.

In fact, the new religious awareness of the relationship between humanity and nature does not happen in un uncontested manner. For instance, in the opposite camp, a strong anthropogenic climate change skepticism has been exerting a sizeable influence in the American political debate, especially since President Trump's election in 2016.

In a number of religious communities one may find the persistence of a dominionist reading of the Scriptures, highlighting, in particular, the belief that humankind has a duty to "fill and subdue the earth" and "turn the wilderness into garden". This perspective has been brought into the public sphere by American evangelicals "striving to inform the environmental policy and frustrate endeavors design to address human-made climate change" (Ronan 2017, p. 2). Evangelicals of this orientation show evidence of "growing climate change denial, the rise of anti-science rhetoric and the politicization of global warming disbelief" (Ronan 2017, p. 2).

In this context, Pope's Francis visit to the United States (22–27 September 2015)—a country where he never had been before his election, unlike former Popes, such as Eugenio Pacelli, Giovanni Battista Montini, Karol Wojtyla and Joseph Ratzinger—was characterized by the public discourse of the new Catholic "political ecology". Pope Francis not only reaffirmed the issues central to the Encyclical, but also and above all underlined the need to rediscover the fundamentals of coexistence and development. He thus opened the doors to the possibility of a more responsible and inclusive economic model that—as he said in front of the Assembly General of the United Nations, which was discussing the "Agenda 2030 for Sustainable Development"—assures everyone a house, a dignified and duly paid job, adequate nutrition and drinking water. Interestingly, Bergoglio's "narration" about global disorder met with a problematic reception in the US, perhaps because he described this disorder as having been caused, among other things, by financial speculation and by the logic of profit, not to mention the damage inflicted to the environment (Ferrara 2016).

## 4. Climate Change and International Relations

Environmental issues and climate change were not the core business of the study of world politics and International Relations Theory in the decades following World War II. The growing salience of global environmental politics has produced only with some considerable delay a genuine interest among academics "in understanding how a loose set of environmental ideas originating in the 19th century came to redefine international legitimacy and the moral purpose of the state in the late 20th century" (Falkner and Buzan 2017, p. 132).

Scholars at the time were not attracted by the subject for a number of reasons: politically, because climate change did not appear to be a structural factor in world politics comparable, for instance, to nuclear proliferation; academically, because they were trying to deal with the causes of the climatic disorder, rather than with the effects. Moreover, International Relations Theory seemed to be insufficiently equipped, from an epistemological perspective, to incorporate in its research agenda the topic of climate change insofar as it was perceived as belonging mostly to the domain of "hard science". In recent decades, however, the process of articulating a green theory of international relations has proceeded at an accelerating pace, with the emergence of new approaches that appear much broader than standard environmentalism.

Many strands of research by now fall within this wide spectrum of studies, regarding

> regimes of international governance at both local and global levels, a variety of issues in international political economy, including development and economic growth, the nature of security, the role of state sovereignty and, and the most basic level, how the problems and challenges generated by environmental degradation are to be conceptualized and theorized. (Lawson 2015, p. 219)

Two main theoretical perspectives have emerged in the context of "green theory": "environmentalism" (with a problem-solving orientation) and "ecologism" (implying a multi-dimensional vision of the environmental problems, requiring a radical change in the modes of production and habits of consumption). Another way of presenting the two academic approaches consists in labelling the former as "modernist" (progress in technology and will generate the knowledge needed to protect the environment) and the latter as "eco-radical" (drastic changes in life-style and the population control are the only ways to ensure sustainable development) (Jacob and Sørensen 2010, p. 263).

At some point, several political scientists became interested in the implications, for national and global governance, of the "Gaia hypothesis".

Back in 1979, James Lovelock elaborated a fascinating and controversial hypothesis of the reduction (or reversal) of entropy as a general characteristic of any form of life. For Lovelock:

> life is a member of the class of phenomena which are open or continuous systems able to decrease their internal entropy at the expense of substances or free energy taken in from the environment and subsequently rejected in a degraded form. (Lovelock 1979, p. 4)

A simplified version of this definition states that "life is one of those processes which are found whenever there is an abundant flow of energy" (Lovelock 1979, p. 4). Lovelock takes a step forward, and contends that:

> some of the activity associated with concentrated entropy reduction within a living system might spill over into conveyor-belt regions and alter their composition. The atmosphere of a life-bearing planet would thus become recognizably different from that of a dead planet. (Lovelock 1979, p. 5)

Based on scientific observations and research hypothesis, Lovelock reaches the conclusion that:

> the only feasible explanation of the Earth's highly improbable atmosphere was that it was being manipulated on a day-to-day basis from the surface, and that the manipulator was life itself. (Lovelock 1979, p. 6)

This is the essential definition of "Gaia", which Lovelock elaborates further by stating that it consists, in sum, in

> the hypothesis that the entire range of living matter on Earth, from whales to viruses, and from oaks to algae, could be regarded as constituting a single living entity, capable of manipulating the Earth's atmosphere to suit its overall needs and endowed with faculties and powers far beyond those of its constituent parts. (Lovelock 1979, p. 9)

This implies, in turn, that the atmosphere "is not a merely biological product, but more probably a biological construction" (Lovelock 1979, p. 9), a sort of "extension of a living system designed to maintain a chosen environment" (Lovelock 1979, p. 9). For Lovelock, "Gaia [is] a complex entity involving the Earth's biosphere, atmosphere, oceans, and soil" (Lovelock 1979, p. 10).

In a sense, the Gaia approach represents a major shift from the world-system mindset to the new intuition of a whole-planet system. In order to be in accordance with the self-regulating mechanisms of the planet, some thinkers (Madron and Jopling 2003) suggest that a new form of political system would be necessary, a "Gaian democracy". This innovative and somewhat normative idea has been described—simply but substantially—as follows:

> Imagine a democratic world as complex, adaptive and flexible as the ecosystems with which it interacts. It is made up of millions of engaged, active citizens connected together in a global network of democracies that transcend the nation-state, all organised around the twin

goals of sustainability and global/social justice. Imagine further that the internal principles of the network are decentralisation, maximum diversity and 'people-power', which together operate as the organising principles of society characterised as a form of 'social learning'. (Barry 2004)

The model of Gaian democracy has been developed in the context of Global Studies, a sub-field—or, better, a trans-field—that does not respond necessarily to the epistemology of International Relations. For that matter, only in recent years has the challenge of climate change and its implications for international politics become a substantial topic among the "traditional" academic disciplines dealing with world affairs. This interest has two different aspects: first, in terms of "diagnostics," it has implied a better understanding of the interconnection between biophysical and political aspects of climate change, leading to the conclusion that the increasing impact of human activities on the bio-sphere represents a complex and urgent multilevel governance challenge; second, it has raised academic awareness on the possibility that this topic may represent a new research agenda for International Relations with potential consequences on the ontology and epistemology of the entire field (Underdal 2017). So far, the issue of ecology in world politics has been considered salient mainly for its practical implications for development, security, health, food supply and multilateralism. The diplomacy of climate change has raised hopes, but it has also generated frustrations among the actors and agents involved. One reason for the slow progress on this topic of global governance has been classical diplomacy's tendency to frame the issue in the context of inter-state relations, implying a burdensome process of finding consensus understood as a global compromise at the level of the lowest common denominator (Elliott 2013, p. 850). Moreover, the negotiations have been affected by the dualism between developed and developing countries on the concrete implications of the principle of "common but differentiated responsibilities" to contain climate change on a global scale (Elliott 2013, p. 850).

After the adoption of the United Nations Framework Convention on Climate Change (UNFCCC) in 1992, a series of conferences (COP—Conference of the Parties) were convened in order to seek a global consensus. These conferences—after limited results, such as the 1997 Kyoto Protocol set up to reduce the emission of "greenhouse gas" (GHG)—coalesced in the COP21 held in Paris in 2015. The reflection note accompanying the Paris Agreement sounds ambitious and perhaps wishful-thinking, since it states that the parties "reached an ambitious agreement, a binding agreement, a universal agreement", adding that "the Paris Agreement is a balanced compromise among all Parties: all made concessions; all received reassurances" (PCOP21 2016). At any rate, participation at the Conference was symptomatic of what had become truly global concern on climate disruption: attendees included representatives of 197 Parties to the Convention, as well as representatives of two observer States, 56 United Nations bodies and programs, convention secretariats, specialized agencies and institutions, and related organizations of the United Nations system. The COP21 in Paris was also attended by representatives of 71 IGOs (inter-governmental organizations) and representatives of 1109 NGOs (non-governmental organizations). It can thus be described as one of the most open, inclusive, participatory and democratic negotiations in recent history. In terms of the level of transparency and inclusiveness it might compete with the negotiation leading to the UN Sustainable Development Goals adopted in 2015. Climate diplomacy (Elliott 2013) may therefore be considered, according to the different analytical perspectives, to be an example of network diplomacy (Heine 2006), multi-stakeholders diplomacy (Kurbalija and Katrandjiev 2006), collective diplomacy (Haas 2008) or public multilateral diplomacy (Bäckstrand 2008). However, in the complex realm of climate diplomacy the stress on an inclusive multilateralism (at least in procedural terms) often produces the unintended consequence of reducing the effectiveness of the process. This explains why, since its inception at the beginning of the 1990s, climate diplomacy has been characterized by the two contradictory features of alarming urgency and frustrating incrementalism (Elliott 2013, p. 842).

In Paris the Parties committed themselves to pursuing the specific objective of holding temperatures "well below" 2 °C (operational goal) while also making efforts to stay below 1.5 °C (aspiration). Parties sought to mitigate the effects of the GHG (Greenhouse Gases) by establishing and making transparent

their national determined contributions (NDCs). Even less stringent are the articles of the Paris Agreement regarding adaptation, "namely to enhance adaptive capacity, strengthen resilience and reduce vulnerability to climate change in the context of the temperature goal" (Bodle et al. 2016, p. 11). The Paris Agreement establishes, taking into account both the asymmetries of world economy and the need for a shared and universal commitment on climate change, the principle of "common but differentiated responsibilities". In connection to the adaption goals, the Paris Agreement also recognizes the loss and damage caused by climate change, even though that was one of the most sensitive questions "due to concerns by developed countries that this could entail state responsibility, liability and claims for compensation" (Bodle et al. 2016, p. 12).

## 5. A "Sublime Communion": Between Communitarianism and Cosmopolitanism

The LS assumes as a truth that "all of us are linked by unseen bonds and together form a kind of universal family, a sublime communion" (LS, §89).

Now, the notion of communion is familiar in the Catholic and Christian discourse, but totally uncommon in the analysis of world affairs, therefore there is a need to find a way to understand it in the context of current international relations. One way to make sense politically of the concept of universal communion is by trying to highlight what it doesn't mean.

To begin with, one statement that is repeated over and over in the Encyclical is that everything is connected. Now, there are different ways of reading this conception. It has little to do either with connectivity (the fact of a world increasingly being linked together by the world wide web, a condition that might lead humanity to a sort of "mental pollution" due to "a mere accumulation of data which eventually leads to overload and confusion"—LS, §47) or with connectography, the innovative research by Parag Khanna about the "emerging global network civilization"(Khanna 2016), where the growing interconnection between mega-cities is more relevant than state borders. This assertion also absolves the famous, metaphorical and innocent Chinese butterfly of chaos theory, which is too often unfairly accused of causing tornados in Texas just by flapping her tiny wings.

The interconnection outlined in the LS has to do with a relational narrative rather than being a reference to an objective connection, as in Ulrich Beck's discourse about "cosmopolitan realpolitik" (Beck 2008).

Moreover, the idea of "connection" diverges substantially from the systemic approach to the study of international relations. The notion of system and structural-realist theory at the core of International Relations was one of the outstanding intellectual achievements of Kenneth Waltz (Waltz 1979). In a quite different meaning, the concept of world system has been one of the main paradigms of the International Political Economy. In particular, Immanuel Wallerstein used this notion to explain the asymmetries and shortcoming of the global capitalist economy. His analysis of the world system resulted in his prediction of political and economic instability, social unrest and, finally, a world-wide economic crisis that will cause the collapse of the system (Wallerstein 1974). For a plausible semantical appraisal of the concept of "connection" it is necessary to read another crucial statement of the Encyclical: that "everything is related" (LS, §92). The notion of relation clarifies the idea of connection in terms of love and respect: "Because all creatures are connected—notes Pope Francis—each must be cherished with love and respect, for all of us as living creatures are dependent on one another" (LS, §42).

This idea of humanity interacting in solidarity with non-human living beings is, of course, a concept wider than the famous Kantian formulation of cosmopolitanism, as expressed in its project for Perpetual Peace: "a violation of rights in one part of the world is felt everywhere" (Reiss 1991, p. 107). Benedict XVI took this aphorism to the next level: "Every violation of solidarity and civic friendship harms the environment" (Benedict XVI 2009, §51).

Diplomatic jargon—including the case of climate diplomacy (Elliott 2013)—has seen the adjective "comprehensive" used in a plethora of political declarations. Indeed, the incredibly high recurrence of "comprehensive" in international documents is beyond . . . comprehension.

Contemporary ecological thought has made progress especially in underlining the inter-connection of life on the planet, but is has been less specific on the linkage between nature and society.

To this regard, the inter-connection outlined in the Encyclical is truly holistic, multidimensional, and systematically violates all the rigorous academic epistemological tenets of the disciplines, fields and subfields, bringing together in the same context science, technology, anthropology, moral theology, international relations, global economy.

The interplay of the diverse factors of human life and natural environment on the planet entails a new kind of ecological thinking, which could be defined, rather than integral ecology, "communional ecology".

The notion of communion is at the crossroads between communitarianism and cosmopolitanism. Ironically, it seems closer to what Andrew Dobson calls "thick cosmopolitanism" (Dobson 2006), since it implies a set of strong moral obligations rather than a generic sense of belonging to a loose "human community". Usually cosmopolitanism is associated with a thin universalism, covering relations between all human beings ("common humanity"). However, this kind of cosmopolitanism lacks one fundamental property capable of triggering actual engagement, namely motivation, which is instead one of the defining features of communitarianism. This lack creates the relative "isolation" in which we find ourselves when facing real world issues, such as extreme poverty in distant places. Therefore, engagement in a cosmopolitan context requires more "nearness . . . to vulnerable, suffering, disadvantaged others" (Dobson 2006, p. 171). In order to operationalize the "nearness" to unknown (and sometimes unreachable) persons, it is essential—according to Andrew Linklater—to examine two very different mechanisms: the first has to do with "emotional dispositions" and empathy; the second consists of a much stronger model, that of the "cosmopolitan emotions" that are likely to surface when the actors become aware that they are causally responsible (even indirectly) for harming others and their physical environment (Linklater 2006; Dobson 2006).

> Causal responsibility produces a thicker connection between people than appeals to membership of common humanity, and it also takes us more obviously out of the territory of beneficence and into the realm of justice. (Dobson 2006, p. 172)

## 6. A Short Walk in the Anthropocenic Woods

The idea of integral ecology implies a profound hermeneutics of the crisis: "we are faced not with two separates crises, one environmental and the other social, but rather with one complex crisis which is both social and environmental" (LS, §104).

The emphasis on interconnection, beyond the search for causal responsibility of human behavior towards nature, underlines "how inseparable the bond is between concern for nature, justice for the poor, commitment to society, and interior peace" (LS, §10). In particular, Pope Francis points to the "intimate relationship between the poor and the fragility of the planet" (LS, §16). Therefore, it appears commonsensical that "a true ecological approach always becomes a social approach; it must integrate questions of justice in debates on the environment, so as to hear both the cry of the earth and the cry of the poor" (LS, §49). This holistic view of the ecological crisis includes, for instance, respect for human dignity and the need "to take account of the value of labour" (LS, §124) with the awareness that a balanced economic development entails the broader objective of allowing everyone a dignified life through work (LS, n. 128).

This peculiar and broader reference to "one complex crisis" rather than a "dual crisis" could be easily inscribed into the same semantic field of the notion of the Anthropocene, defined by Paul Crutzen as a new geological epoch of the world characterized by the pervasive influence of human behavior on the entire planet, to the point that the transformation even of the basic physical features of the globe, in sharp contrast with past eras, are induced and accelerated by human activities (Crutzen and Stoermer 2000). It is the present structure of human relations and the mode of production and consumption that interfere heavily with the state and the destiny of Earth well beyond the human generations. In fact,

without major catastrophes like an enormous volcanic eruption, an unexpected epidemic, a large-scale nuclear war, an asteroid impact, a new ice age, or continued plundering of Earth's resources by partially still primitive technology (the last four dangers can, however, be prevented in a real functioning noösphere) mankind will remain a major geological force for many millennia, maybe millions of years, to come. To develop a world-wide accepted strategy leading to sustainability of ecosystems against human induced stresses will be one of the great future tasks of mankind, requiring intensive research efforts and wise application of the knowledge thus acquired in the noösphere, better known as knowledge or information society. (Crutzen and Stoermer 2000, p. 18)

Another way of looking at the Anthropocene hypothesis is to assume that it has become difficult to distinguish between societies and environments, "since they have merged into a complex socionatural entanglement" (Arias-Maldonado 2016, p. 10), as a result of a process of "human colonization" producing an "hybridization by which nature is gradually losing its autonomy from society". Therefore, "it makes sense to see the Anthropocene as culminating The Great Hybridization of society and nature" (Arias-Maldonado 2016, p. 9).

In a slightly different interpretation, the Anthropocene indicates:

an emerging epoch in which human influences become decisive in affecting the parameters of the Earth system, accompanied by the potential to generate instability and even catastrophic shifts in the character of the whose system ("state shifts") of the sort that are common in the planet's deeper history, but unknown in recorded human history. (Dryzek and Pickering 2019, p. 2)

The Anthropocene also alters the ongoing conceptualization of physical forces, introducing into the human understanding of the planet the nonlinear dynamism of the complexity sciences. Furthermore, it changes the notion of time, by situating humanity within a geological scale, and interpreting the contemporary industrial and technological order "as a moment in a long history of human-energy-agricultural-political configurations" (Allan 2018, p. 279). Lastly, the appearance of the Anthropocene transforms the perception of humanity as one species among others, since it has become a geological force with the diverging effects of engendering humility (however, still keeping an impulse of dominance) or, on the contrary, generating a sense of impotence vis-à-vis the unpredictable turns of the Earth system (Allan 2018, p. 280).

Against this backdrop, the Pope's reading of the current state of the world is consistent with the new appraisal of the position of humanity over the long history of planet Earth. There are two conceptual lens through which our present global human condition is interpreted: the idea of care, and, by implication, the notion of responsibility. These two practices should ideally replace the search for dominance and hegemony, and they seem to be the "new realism" of "anthropocenic" politics.

Interestingly, Pope Francis doesn't use the biblical narrative of the "garden" entrusted by God to mankind to frame the relationship between humanity and the Earth. He prefers a different image, that of a "common home". It is the original meaning of the Greek word "oikos" (present in the modern concepts of economy and ecology). In some interpretations, the notion of a common home is more inclusive than the idea of creation, the latter being more closely linked to Christianity. Moreover, common home seems to refer to something broader than the environment, implying in fact a strong connection between creation and society (Stefani 2015). It is a less anthropocentric (bio-centrist?) form of Christianity, a Christianity which is readier to experience "fraternity" with the whole of creation. Pope Francis' insistence on connections includes interpersonal, social, ecological, and even cosmological bonds (Morandini 2015). This leads to a theological reformulation of the notion of "eco-system": "each organism, as a creature of God, is good and admirable in itself; the same is true of the harmonious ensemble of organisms existing in a defined space and functioning as a system. Although we are often not aware of it, we depend on these larger systems for our own existence" (LS, §139).

In fact, contemporary ecological politics stresses that

> human beings are embedded—not so much in the communitarian sense that we derive
> our norms and beliefs from specific cultural contexts, but in the embodied sense that we
> are organisms whose production and reproduction depend on the adequate provision of
> environmental goods and services. This metabolistic relationship with our non-human
> natural environment constitutes the ineluctable frame within which our political projects are
> carried out. (Dobson 2006, p. 176)

At this point, it should be acknowledged that "care for the common home" is a potent metaphor,
as long as it replaces the paradigm of human hegemonic power. Quoting the philosopher Romano
Guardini, the Encyclical states that in our technocratic world "power is never considered in terms of
responsibility of choice which is inherent in freedom" since "its only norms are taken from alleged
necessity, from either utility or security" (Guardini 1998, p. 82).

In this precise context of mutual responsibility we should also understand the principle of the
climate as a common good, "belonging to all and meant for all. At the global level, it is a complex
system linked to many of the essential conditions for human life" (LS, n. 23), acknowledged by the
wider international community. However, there is still no consensus in the international system on how
to make sense of the notion of "common good" by dealing in an incisive way with the management of
the global commons, through an effective, inclusive and consensual system of governance.

This lack of consensus is also due to the fact that the typology of the "common goods" could be
broad and profoundly diverse. One first essential distinction to be made when dealing with common
goods in terms of natural resources, is between natural stocks and funds. Whereas stocks are consumed
after being used (as in the cases of coal and oil) and appear in the form of flows, funds refer to living
(plants, animals) and non-living nature (air, water, sun) and take the feature of service provision
(Arias-Maldonado 2016). Another important differentiation regards disposable (irrelevant), fungible
(important but not critical) and critical (irreplaceable) natural capital.

In terms of climate change, the set of international agreements and conventions can be included in
the category of "regulatory" collective goods, as distinct from productive, distributive and redistributive
collective goods (Cerny 2010, p. 100).

In this respect, the present debate of climate change as a "governing-the commons-problems"
should not consider the atmosphere in the context of the traditional attributions of the common
resources, usually defined by the two qualities of being nonexcludable and nonrivalrous. In fact,

> Any solution to governing this commons require the recognition that our emissions are
> actually rivalrous from the point of view of achieving any global limit, and so they require
> some form of regulation to be made compatible. (Lane 2016, pp. 110–11)

One promising way to operationalize politically the idea of common good related to the
environment is through the concept and practice of energy democracy. Energy democracy:

> means that everybody is ensured access to sufficient energy. Energy production must thereby
> neither pollute the environment nor harm people. More concretely, this means that fossil
> fuel resources must be left in the ground, the means of production need to be socialized and
> democratized, and that we must rethink our overall attitude towards energy consumption.
> (Angel 2016, p. 10)

In the same vein, it is worth mentioning the "Green New Deal", a US based political platform that
claims to "convert the old, gray economy into a new, sustainable economy that is environmentally
sound, economically viable and socially responsible." The platform demands, among other things, to
"enact energy democracy based on public, community and worker ownership of our energy system.
Treat energy as a human right (Green Party US 2019)".

There is no doubt that the mode of production and distribution of energy at state and international level has heavy consequences on the models of governance and the concentration/diffusion of power. There are, however, two opposing interpretations of the political and institutional aspects of the transitions towards clean and renewable energy. On the one hand, solar energy is said to be deeply connected with empowerment and self-government of local communities, enabling them to become less dependent on the choices of national and trans-national corporations pushing towards concentrating decision-making processes at the top of the economic and political scale. In this reading, energy democracy is the opposite of energy authoritarianism. On the other hand, there is an alternative way of reading the global transition towards a green and circular economy: in this version, the process needs a robust political and institutional drive, since huge investments and R and D initiatives are needed in order to make renewable energy both accessible and affordable for all. Contrary to the narrative of decentralizing energy power, this understanding of energy transition implies, for instance, strong governments, credible international institutions, cogent rules and regulations, especially in order to create incentives and inducements.

## 7. Climate Justice and Moral Agency

Still, the main practical question for international relations remains "who should care, and for what?" Actually, one of the most debated issues in international relations is the relationship between moral agency and responsibility. Who is responsible for what in a plethora of international relevant actors, such as nation states, international organizations, transnational corporations, non-governmental organizations, big decision-makers in the global economy? When it comes to moral agency, the normative approach to international relations insists on duties and obligations ("something must be done"), whereas international history elaborates on blame and accountability ("never again") (Erskine 2008). However, in both cases, failure to indicate the moral agents involved makes such refrains meaningless. The search for what types of collectivity at the international level (states, international organizations) could be charged with moral responsibility for humanitarian or environmental crises, under which conditions and to what effects remains a fundamental analytical task (Erskine 2008, p. 704).

In the present controversy on climate change, the debate is not so different from other domains, like financial speculations and poverty, or selling weapons in relation to the spreading of conflicts and instability. In particular, the main questions for environmental ethics are the following:

> Is human conduct in relation to natural ecosystems properly subject to moral constraints, or are they applicable only to ways humans treat each others? If the answer is yes, what particular moral constraints are involved, and how are they different from those governing our actions towards other humans? How would the standards and rules arising from those constraints be rationally justified?. (Lawson 2015, p. 234)

In the LS the problem of global responsibility is raised in a direct and explicit way. The "ethics of international relations" implies the acknowledgment of differentiated responsibilities (LS, §52) toward the poor, the weak and the vulnerable. It combines normative and empirical evaluations of the asymmetrical conditions between states and other international actors. These are initial elements for an International Political Economy of climate change, first outlined by Pope Francis' predecessor, Benedict XVI, when he advocated in favor of "eliminating the structural causes of the dysfunction of the world economy and correcting models of growth which have proved incapable of ensuring respect for the environment" (Benedict XVI 2007).

However, the main focus of responsibility in the Anthropocene is on a wider and deeper concept of justice: justice beyond national borders (escaping state-centric accounts of justice); justice for non-humans ("other living organisms as well as ecosystems that comprise both living and non-living things") (Dryzek and Pickering 2019, p. 71); justice for non-humans; justice across generations (future human beings living on the planet).

These three perspectives on justice characterize the intellectual transition from the idea of environmental justice to the notion of planetary justice (Dryzek and Pickering 2019, pp. 68–73). As for the location of the agency in the Anthropocene, the individualist-collectivist divide in terms of responsibility might be bridged through a process of "ecological reflexivity", meaning "the capacity of humans to engage in political contestation about what justice should mean in changing conditions" (Dryzek and Pickering 2019, p. 79). In this framework, the content of planetary justice would imply "viewing humanity's relationship with the Earth system as central to planetary justice, rather than keeping justice in splendid isolation from the Earth system" (Dryzek and Pickering 2019, p. 80).

The reference to the "environment" in the LS actually implies "a relationship existing between nature and the society which lives in it. Nature cannot be regarded as something separate from ourselves or as a mere setting in which we live. We are part of nature, included in it and thus in constant interaction with it" (LS, §139).

## 8. International Politics as Bio-Politics

In terms of the search for transnational and planetary justice, the effects on global, communal and human security of the current environmental crisis are of paramount importance. The very concept of security is undergoing a process of restructuring in the light of climate change. However, there are at least two different paths to rethinking security in the Anthropocene.

The first and most common way to proceed towards such a re-articulation is in terms of environmental security, understood as a potential violent struggle over scarce natural resources (Fagan 2017, p. 295). This approach is the main option of government and public agencies when they connect security and climate change, and it is based on the ongoing dualism human/nature. This mental process doesn't require rethinking the very concept of security itself, limiting the reframing to an incorporation of environmental issues into the traditional security framework.

Interesting strands of research on climate change trace the relations between climate insecurity, human security and conflict. For instance, desertification affects large areas of the globe, seriously affecting the livelihood of millions of human beings. Interstate or intrastate conflicts may thus erupt due to the scarcity of natural resources and the structural change in terms of food production and water availability (Barnett and Adger 2007). A paper presented in 2007 by Javier Solana Madariaga, then EU High Representative for the Common Foreign and Security Policy, states that:

> climate change is best viewed as a threat multiplier which exacerbates existing trends, tensions and instability. The core challenge is that climate change threatens to overburden states and regions which are already fragile and conflict prone. It is important to recognize that the risks are not just of a humanitarian nature; they also include political and security risks that directly affect European interests. (Solana and the European Commission 2008)

The main critical consequences of climate change are identified by Solana in conflict over resources, economic damage and risk to coastal cities and critical infrastructure, loss of territory and border disputes, environmentally-induced migration, situation of fragility and radicalization, tension over energy supply, pressure on international governance.

Pope Francis has a clear understanding of the complexity of contemporary conflicts, which are often triggered by phenomena apparently far from the international political context, such as the degradation of the environment. "It is foreseeable that, once certain resources have been depleted, the scene will be set for new wars, albeit under the guise of noble claims. War always does grave harm to the environment and to the cultural riches of peoples, risks which are magnified when one considers nuclear arms and biological weapons" (LS, §57).

This is, for example, the complex but very concrete theme of the so-called "ecowars": conflicts and tensions that are related to climate change, considered by analysts as a real "threat multiplier" for global security (Rüttinger et al. 2015).

More broadly, there are multiple ways and mechanisms by which climate change may affect human security, implying structural challenges on vulnerable livelihood (hitting populations with high-resource dependence), poverty (insofar as climate change may increase relative deprivation), weak states (undermining government capacities to provide essential services and resources), migration (as an additional "push" factor) (Barnett and Adger 2007, p. 663).

Among the most relevant environmental factors potentially causing tensions and conflicts is the control of waterways and access to water resources. It is estimated that around 1.2 billion people are already living in areas affected by water stress (UN Water 2015). Hydro-crises may push populations to move from areas hit by rising sea levels, storms, and floods, leaving agricultural lands to become too dry to cultivate. This may cause massive internal migrations towards urban areas, which are already suffering due to the concentration of the population in the megalopolis, with huge consequences on services and infrastructure, while cross-border migrations increase the probability of conflict over land and access to scarce resources. In general, climate change increases social, economic, institutional and political "fragility", with major effects on stability, cooperation and peace.

The debate on the consequences of climate change or climate chaos on global security has additional theoretical and practical implications. In particular, it shows how the classic distinction between high politics (related to the boundaries of the polity, to security and economy) and low politics (transportation, environment, development aid, education) has been blurring dramatically in recent decades. One example may epitomize this transformation: the division of competences between the G8 (now G7) and the G20. It is very difficult to state that the G7 deals with high politics whereas the G20 deals with low politics. No politics is higher, today, than the one dealing with the future of the planet; it entails dramatic problems of security and prosperity.

The second way to think about security in the Anthropocene consists in building on the broader concept of ecological security by "focusing on the close ties between the human and non-human world, tracing the implications of an understanding of the world in terms of the complex interdependence of ecosystems" (Fagan 2017, p. 300). This represents a further step towards a new conception of security in the condition of the Anthropocene; however, the real paradigm shift would be represented by a rethinking of politics itself "as something other than security politics, as a politics if vulnerability" (Fagan 2017, p. 311).

In more general terms, international politics today is always international bio-politics. This is the case, for instance, with the stockpiles of nuclear weapons capable of destroying the entire planet several times. It is also the case of the lack of food security and access to clean water in many areas of the globe. It is the case for bio-chemical weapons. But it is also the case for the increasing loss of bio-diversity and obviously for the severe health damages caused by pollution: "exposure to atmospheric pollutants produces a broad spectrum of health hazards, especially for the poor, and causes millions of premature deaths" (LS, §20).

Following David Joseph Wellman, what is clearly urgent is an analytical transition from the notion of national survival, central to Realist interpretation of international relations, to eco-Realism, which latter claims that the real goal "is bioregional and ultimately global survival" and that "true self-interest must always be grounded in mutual interest" (Wellman 2010, p. 29). This transition is part of a wider trend of the international system of states from classical concerns of war, balance of power, and order, towards "an expanding agenda of shared-fate issues such as weapons of mass destruction, transnational terrorism, cyber security, migration and the management of the global economy" (Falkner and Buzan 2017, p. 150).

## 9. A Global Conversation

Environmentalism, which started as an advocacy activity in the international arena, seems to have gone through a substantive process of internationalization, to the point that can be considered part of the "primary institutions" of international relations, understood as "deep and relatively durable social practices in the sense of being evolved more that designed" (Buzan 2014). By contrast, "secondary

institutions" are deliberately created, taking the form of inter-governmental organizations or the wider shape of international regimes. The approach of the LS seems to build on environmentalism as a primary institution, with the same level of structural salience as international security (world peace) or global justice (development). In general, environmentalism, as a primary institution of international relations, is slowly moving from a pluralist logic of coexistence (balance of power) towards a solidarist logic of cooperation, or convergence, around shared values (such as human rights, market, development) (Buzan 2004).

These aspects of the climate crisis regarding global governance are commented extensively in Chapter V of the Encyclical.

The Encyclical's plea is "for a new dialogue about how we are shaping the future of our planet. In fact, we need a conversation which includes everyone, since the environmental challenge we are undergoing, and its human roots, concern and affect all of us" (LS, §14). This is an important point. International relations are inherently dialogical and can be compared to an ongoing, permanent conversation that is inclusive and undertaken on an equal footing. There is something of Habermas' communicative-dialogical approach to the social interaction combined with a teleological perspective on improving the common understanding of the actions needed to save the environment. An extended dialogue is always necessary to reach an agreement in international negotiation. As Avishai Margalit puts it,

> on the whole, political compromises are a good thing. Political compromises for the sake of peace are a very good thing. Shabby, shady and shoddy compromises are bad but not sufficiently bad to be always avoided at all costs, especially not when they are concluded for the sake of peace. Only rotten compromises are bad enough to be avoided at all costs. But then, rotten compromises are a mere tiny subset of the large set of possible political compromises. (Margalit 2010, p. 16)

What is interesting here is the not-so-nuanced distinction between "bad deals" and "rotten compromise". In the first case, "bad" stands for the insufficient protection of one's interests (as perceived unilaterally). "Rotten compromise", on the contrary, refers to inacceptable concessions in terms of morality (for instance, in terms of protection of human rights). Perhaps the Paris agreement on climate change of 2015 is not as historic, durable and ambitious as it was hailed to be at the time. Nevertheless, it is really difficult to think that the agreement, involving the entire UN membership, reached at the culmination of more than 23 years of international attempts to forge collective action on climate, "marked by discord and failure, the refusal of the biggest emitters to take part, ineffective agreements and ignored treaties" (Harvey 2015) is either a bad deal or, worse, a rotten compromise. The Paris agreement, technically speaking, is a just agreement in terms of the process of negotiation and an acceptable compromise in terms of substance (environmental justice). That is why it is difficult to understand the rationale, besides political opportunism and strict self-interest, of the decision taken by the United States to cease the implementation of the Paris Accord (correctly defined . . . . "non-binding"!). In his statement on 1 June 2017, President Trump said that "the Paris Accord would undermine our economy, hamstring our workers, weaken our sovereignty, impose unacceptable legal risks, and put us at a permanent disadvantage to the other countries of the world" (Trump 2017) announcing at the same time unilaterally the intention to re-negotiate a multilateral treaty, one of the most complex in the history of diplomacy.

## 10. Conclusions: Towards Sustainable International Relations

In order to overcome the multifaceted socio-ecological crisis, Pope Francis suggests that "what is needed is a politics which is far-sighted and capable of a new, integral and interdisciplinary approach to handling the different aspects of the crisis" (LS, §197).

Some concrete steps should be taken in the direction of a more sustainable international socio-economic and political order.

At horizontal level, one ambitious goal is to bring about a "sustainable democracy" by fostering a transnational "ecological citizenship" (LS, §211) that would transform every-day material life in order to reduce patterns of compulsive consumerism.

An important contribution may stem from religious communities. As for the Christians, the Pope advocates for an "ecological conversion" (LS, §217), by calling for a number of virtuous attitudes, in particular "generous care" (LS, §220), as an expressions of "civic and political love". This applies also to other global religions, since "the majority of people living on our planet profess to be believers. This should spur religions to dialogue among themselves for the sake of protecting nature, defending the poor, and building networks of respect and fraternity" (LS, §201). Building on the main pillars of the Encyclical, there has been an interesting attempt to articulate the "Ten Green Commandments" in order to reflect both the analytical and ethical perspectives of the Encyclical: take care of our common home in peril; listen to the cry of the poor; rediscover a theological vision of the natural world; recognize that the abuse of creation is ecological sin; acknowledge the human roots of the crisis of our common home; develop an integral ecology; learn a new way of dwelling in our common home; educate toward ecological citizenship; embrace an ecological spirituality; cultivate ecological virtues (Kureethadam 2019).

An equally radical approach elaborates on the individual responsibility vis-à-vis the environment and the anthropogenic climate, highlighting the problematic aspects of private property from the vantage point of Monotheistic traditions, with a focus on the nature of the human person, the community and obligation (Babie 2016).

An additional asset is the epistemological dialogue among the various sciences in order to better understand and interpret the different elements of the climate crisis.

Last but not least, a collaboration beyond contrasting ideologies is required between the diverse ecological movements.

To put it shortly, the Encyclical considers essential the construction of a transnational and transcultural network that could serve as the infrastructure of ecological change in world politics.

That would not be sufficient unless there is a parallel "mental transition" from the old way of thinking (pathological path-dependency within institutions, that may represents a formidable obstacle to change) to a new mindset, an Anthropocenic reflexivity understood as "the capacity of structures, systems, and sets of ideas to question their own core commitments, and if necessary change themselves; to be something different, rather than just do different things" (Dryzek and Pickering 2019, p. 17).

Against the backdrop of this structural transition, it is important to re-frame the concept of sustainability in the light of reflexivity. Reflexive sustainability should first and foremost escape from the trap of "conservative" sustainability understood as adaptation, since the dominant neo-liberal discourse has already "coopted" such a notion in order to do some "greenwashing" of modes of productions and models of lifestyle on a global scale. A reflexive conception of sustainability should be open (allowing a de-centralized assessment), ecologically grounded (recognizing the interaction between social systems and the Earth system), dynamic (in order to reflect changes in the interactions between humanity and the Earth), far-sighted (strongly orientated towards the future), and integrated (in two senses: "internal" integration, across different sustainability problems; "external" integration between different values, for instance sustainability and justice, or reconceptualizing democracy in Anthropocene) (Dryzek and Pickering 2019, pp. 86–94).

As far as International Relations are concerned, an eco-realistic perspective refuses the traditional agnosticism of "Realism" about what is good and evil in world politics, arguing that "what is good promotes the sustainability of the human and non-human members of the biosphere, and what is destructive is whatever undermines the capacity for building, sharing and maintaining sustainability—domestically and across transnational borders" (Wellman 2010, p. 30). Eco-Realism builds on the concrete concept of "bioregions" as natural geographic and cultural-anthropological entities often artificially divided by national borders.

On a more pragmatic level, international actors should always consider first what is their best alternative to a negotiated agreement. In the case of climate negotiation, the idea that a state would be better off without an agreement is tenable only on the following conditions: the state has enough power to walk out unilaterally without suffering major consequences in the short term; the state is not concerned by the damage to its reputational capital on the world scene, even though it may have effect in the medium-long term; the policy perspective is the short term, since in the long term the effects of climate change would affect also the withdrawing country.

Second, international actors should consider to what extent the preeminence of the alleged national interest is actually *in* the national interest. For instance, it is impossible today to talk about national interest without taking into account intergenerational solidarity as a basic question of justice that considers not only space (actual citizens on a given territory) as the object of political realm, but also time (the future members of the polity). Actually, many analysts doubt the will or ability of national governments to represent the concern of future generations within their territory adequately, and to ensure that issues are settled at their expense" (Albin 2001, p. 28).

The same notion of a foreign policy based on national interest should be checked against two different moral postures. The first version, the Obligatory Exclusivist Thesis, implies that "when the national interest conflicts with other values, the national interest should always take precedence" (Buchanan 2005, p. 111). A second, alternative version of this formulation is the Permissible Exclusivity Thesis, stating that "it is always permissible for a state's foreign policy to be determined exclusively by the national interest" (Buchanan 2005, p. 111). Whereas in the first case the pursuit of the national interest as a supreme goal of foreign policy is mandatory as an invariable moral obligation, in the second case it is a deliberate choice, no less moral in terms of comparative goods for the state (for instance, when the same survival of the state is at stake; however, in the case of climate change, the notion of national survival doesn't hold per se and should be reconsidered in the contexts of the survival of mankind).

Third, the damage that could be done to the "global conversation" required to face global challenges by undermining international efforts should be discounted. It is not sufficient to state that "our planet is a homeland and that humanity is one people living in a common home" (LS, n. 164); what we need is "one world with a common plan". A plan that cannot be imposed by anybody and on anybody; it requires global consensus in order to sign and implement enforceable international agreements, "since local authorities are not always capable of effective interventions. Relations between states must be respectful of each other's sovereignty but must also lay down mutually agreed means of averting regional disasters which eventually affect everyone" (LS, §173).

In sum, a sustainable international politics is a context of mature international relations conducted under the principles of care, responsibility, consensus, justice; a place of encounters, conversation and dialogue for the common good and for the protection and promotion of the global commons. It brings together empirical and normative arguments, as in the case of "deep ecology" or "ecosophy"—a normative worldview developed by the Norwegian philosopher Arne Næss (Næss 2016)—"which emphasizes the intrinsic worth of all being, from microbes to elephants, as well as the respect for cultural diversity, social justice and advocacy of non-violence in all sphere, both natural and cultural" (Lawson 2015, p. 232).

Going back to the initial quotation (from LS) about the unsustainability of the present world system, the reasons are by now adamant: insufficient awareness, in terms of policies, of the embedded condition of humanity in the biosphere; reticence in recognizing that, contrary to the tenets of Realism, hard power cannot control systemic processes taking place on Earth and, as a consequence, the need to re-conceptualize the notion of security; failure to acknowledge the centrality of religious beliefs, practices and traditions as fundamental elements of sustainability understood as a rich basket of complex interactions between social–cultural–political communities beyond national boundaries; resistance toward entirely new patterns of collective behavior beyond the limited scope of mitigation and adaptation in the field of climate change.

Practicing sustainable international relations, by contrast, means exiting the logic of power or hegemony, and embracing also at the global level two wonderful virtues outlined in the LS. The first is healthy humility, requiring that states and international institutions accept and play with dignity, engagement and responsibility an equal and constructive role in world politics instead of seeking dominance (whether military or economic). The second is happy sobriety, requiring that international actors strengthen and support the global effort for a circular and inclusive economy instead of dissipating irreplaceable resources such as land, air, and water and contaminating the planet with fossil fuel. Two concrete steps to build an international "culture of care" (LS, §231) and practicing an attitude of "tenderness" toward both the poor and the earth (LS, *A prayer for our earth*).

**Funding:** This research received no external funding.

**Conflicts of Interest:** The author declares no conflict of interest.

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
