# Peer review of "Sustainable International Relations. Pope Francis’ Encyclical Laudato Si’ and the Planetary Implications of “Integral Ecology”"

_religions, doi:10.3390/rel10080466_

Round 1
Reviewer 1 Report
What I think is particularly creative about the article is its attempt in places to bring together theology and the theory of International Relations as part of our attempt to explain or really understand what is happening in the world. The author clearly does have a IR background and perhaps an English School of IR background (which is great!).
However, in my comments I raised the issue of 'net-idealism' and what I mean is that the environment is only one of a variety of issues in which IR scholars have said - well, 'we have to change' ... and, what happens? Not much, the issues of power, authority, and legitimacy are always there and need to be contended with. Society at large is not going to accept perhaps this view - and the more radical ecological folks - the emphasis on fear, human extinction - what the perspective of this paper perhaps reflects in this pope - is also a sense of hope (Benedict XVI encyclical on hope also talks about the environmental crisis). Going back to St. Francis and the early rise of capitalism - part of his conversion was indeed seeing that the way he lived out his life - the way we live out our lives, affects more people than we know .... theory in this sense as everyday social practice.
Author Response
Dear Reviewer,
Thanks a lot for your comments and suggestions. I have incorporated them in a new version, that I will upload soon. In several parts I have carefully reconsidered the risk of "net-idealism", in order to make sure that my reading of the Encyclical and more generally of the relationship between eco-theology, IR and social practice become more operational. I have also discussed the (international) role of the Holy See in connection with climate change and global religions criticism of power, hegemony, capitalism as a radical interpretation of climate disruption. I hope you will agree. Warmly
Author
Reviewer 2 Report
This article is outstanding. It it is the first such piece of scholarship I have seen that really understands the nexus between religion, the environment, and socio-politico-legal structures, and the demands which what we are doing to the environment make of us to change the way in which we live. Perhaps the only comment I would make is that the author may want to indicate how religion--in the form of the Ten Green Commandments (see https://litpress.org/Products/6363)--and politics--in the form of the Green New Deal (see eg https://www.gp.org/green_new_deal)--may already be responding to the challenge, and to Francis's Encyclical. And, indeed, there is substantial work by Paul Babie on the way in which such considerations must force us to reconsider the operation of the building-block concept of everything this author addresses: private property.
Author Response
Dear Reviewer,
I found your comments and remarks extremely appropriate and stimulating. In particular, I wish to express my gratitude for the relevant reference to the "Ten Green Commandments" , the "Green New Deal" and the important work by Paul Babie. I've included these new perspectives in the paper, since they help understanding well the policy implications of a religious engagement vis-à-vis climate disruption. I will upload the new version of the paper in a couple of days. Warmly,
Author